# Shifted Compression Framework: Generalizations and Improvements

Egor Shulgin[1]

Peter Richtárik[1]

[1]King Abdullah University of Science and Technology (KAUST), Thuwal, Saudi Arabia

## Abstract

Communication is one of the key bottlenecks in the distributed training of large-scale machine learning models, and lossy compression of exchanged information, such as stochastic gradients or models, is one of the most effective instruments to alleviate this issue. Among the most studied compression techniques is the class of unbiased compression operators with variance bounded by a multiple of the square norm of the vector we wish to compress. By design, this variance may remain high, and only diminishes if the input vector approaches zero. However, unless the model being trained is overparameterized, there is no a-priori reason for the vectors we wish to compress to approach zero during the iterations of classical methods such as distributed compressed SGD, which has adverse effects on the convergence speed. Due to this issue, several more elaborate and seemingly very different algorithms have been proposed recently, with the goal of circumventing this issue. These methods are based on the idea of compressing the *difference* between the vector we would normally wish to compress and some auxiliary vector that changes throughout the iterative process. In this work we take a step back, and develop a unified framework for studying such methods, both conceptually and theoretically. Our framework incorporates methods compressing both gradients and models, using unbiased and biased compressors, and sheds light on the construction of the auxiliary vectors. Furthermore, our general framework can lead to the improvement of several existing algorithms, and can produce new algorithms. Finally, we performed several numerical experiments to illustrate and support our theoretical findings.

## 1 INTRODUCTION

We consider the distributed optimization problem

$$\min_{x \in \mathbb{R}^d} \left[ f(x) := \frac{1}{n} \sum_{i=1}^{n} f_i(x) \right], \qquad (\star)$$

where $n$ is the number of workers/clients and $f_i : \mathbb{R}^d \to \mathbb{R}$ is a smooth function representing the loss of the model parametrized by $x \in \mathbb{R}^d$ for data stored on node $i$. This formulation has become very popular in recent years due to the increasing need for training large-scale machine learning models (Goyal et al., 2018).

**Communication bottleneck.** Compute nodes have to exchange information in a distributed learning process. The size of the sent messages (usually gradients or model updates) can be very large, which creates a significant bottleneck (Luo et al., 2018; Peng et al., 2019; Sapio et al., 2021) to the whole training procedure. One of the main practical solutions to this problem is lossy *communication compression* (Seide et al., 2014; Konečný et al., 2016; Alistarh et al., 2017). It suggests applying a (possibly randomized) mapping $\mathcal{C}$ to a vector/matrix/tensor $x$ before it is transmitted in order to produce a less accurate estimate $\mathcal{C}(x) : \mathbb{R}^d \to \mathbb{R}^d$ and thus save bits sent per every communication round.

**Compression operators.** The topic of gradient compression in distributed learning has been studied extensively over the last years from both practical (Xu et al., 2020) and theoretical (Beznosikov et al., 2020; Safaryan et al., 2021c; Albasyoni et al., 2020) approaches. Compression operators are typically divided into two large groups: *unbiased* and *biased* operators. The first group includes methods based on some sort of rounding or *quantization*: Random Dithering (Goodall, 1951; Roberts, 1962), Ternary quantization (Wen et al., 2017), Natural (Horváth et al., 2019a), and Integer (Mishchenko et al., 2022) compression. Another popular example is random *sparsification* – Rand-K (Wangni et al., 2018; Stich et al., 2018; Konečný and Richtárik, 2018), which preserves only a subset of the original vector coordi-

*Accepted for the 38th Conference on Uncertainty in Artificial Intelligence* (UAI 2022).

Table 1: Overview of results for methods obtained as special cases of our general framework DCGD-SHIFT (Alg. 1). Iteration complexities are presented in $\tilde{\mathcal{O}}$-notation to omit $\log 1/\varepsilon$ factors and for the simplified case $\omega_i \equiv \omega, \delta_i \equiv \delta, L_i \equiv L$, $p_i \equiv p$. More refined statements are in theorems with links in the last column. Complexities for DCGD-SHIFT and GDCI are shown in the interpolation regimes: $\nabla f_i(x^\star) = 0 = x^\star - \gamma \nabla f_i(x^\star)$.

| Instance of DCGD-SHIFT | Shift | Previous | Our result | Theorem |
|---|---|---|---|---|
| DCGD-FIXED (this work) | (6) | − | $\kappa\left(1+\frac{\omega}{n}\right)$ | 1 |
| DCGD-STAR (this work) | (8) | − | $\kappa\left(1+\frac{\omega}{n}\left(1-\delta\right)\right)$ | 2 |
| DIANA (Mishchenko et al., 2019) | (10) | $\max\left\{\kappa\left(1+\frac{\omega}{n}\right),\omega\right\}$ | $\max\left\{\kappa\left(1+\frac{\omega}{n}(1-\delta)\right),\omega(1-\delta)\right\}$ | 3 |
| Rand-DIANA (this work) | (12) | − | $\max\left\{\kappa\left(1+\frac{\omega}{n}\left(1-\delta\right)\right),\frac{1}{p}\right\}$ | 4 |
| GDCI (Khaled and Richtárik, 2019) | (13) | $\kappa^2\left(1+\frac{\omega}{n}\right)$ | $\kappa\left(1+\frac{\omega}{n}\right)$ | 5 |

nates. These two approaches can also be combined (Basu et al., 2019) for even more aggressive compression. There are also many other approaches based on low-rank approximation (Vogels et al., 2020; Wang et al., 2018; Safaryan et al., 2021b), vector quantization (Gandikota et al., 2021), etc. The second group of biased compressors mainly includes greedy sparsification – Top-K (Alistarh et al., 2018; Stich et al., 2018) and various sign-based quantization methods (Seide et al., 2014; Bernstein et al., 2018; Safaryan and Richtárik, 2021). For a more complete review of compression operators, one can refer to the surveys by Xu et al. (2020) and Beznosikov et al. (2020); Safaryan et al. (2021c).

**Optimization algorithms.** Compression operators on their own are not sufficient for building a distributed learning system because they always go along with optimization algorithms. Distributed Compressed Gradient Descent (DCGD) (Khirirat et al., 2018) is one of the first theoretically analyzed methods which considered arbitrary unbiased compressors. The issue with DCGD is that it was proven to converge linearly only to a neighborhood of the optimal point with constant step-size. DIANA (Mishchenko et al., 2019) fixed this problem by compressing specially designed gradient differences. Later DIANA was generalized (Condat and Richtárik, 2021), combined with variance reduction (Horváth et al., 2019b), accelerated (Li et al., 2020) in Nesterov's sense (Nesterov, 1983) and by using smoothness matrices (Safaryan et al., 2021a) with a properly designed sparsification technique.

On the other side are methods working with biased compressors, which require the use of the error-feedback (EF) mechanism (Seide et al., 2014; Alistarh et al., 2018; Stich and Karimireddy, 2020). Such algorithms were often considered to be better in practice due to the smaller variance of biased updates (Beznosikov et al., 2020). However, it was recently demonstrated that biased compressors can be incorporated into specially designed unbiased operators,

and show superior to error-feedback results (Horváth and Richtárik, 2021). In addition, error-feedback was recently combined with the DIANA trick (Gorbunov et al., 2020), which led to the first linearly converging method with EF. Later Condat et al. (2022) proposed a unified framework for methods with biased and unbiased compressors.

**Compressed iterates.** Most of the existing literature (including all methods described above) focuses on compression of the gradients, while in applications like Federated Learning (McMahan et al., 2017; Konečný et al., 2016; by: Peter Kairouz and McMahan, 2021), it is vital to reduce the size of the broadcasted model parameters (Reisizadeh et al., 2020). This demand gives rise to optimization algorithms with compressed iterates. The first attempt to analyze such methods was done by Khaled and Richtárik (2019) for Gradient Descent with Compressed iterates (GDCI) in a single node set up. Later GDCI was combined with variance-reduction for noise introduced by compression and generalized to a much more general setting of distributed fixed-point methods (Chraibi et al., 2019).

**Summary of contributions.** The obtained results are summarized in Table 1, with the improvements over previous works highlighted. The main contributions include:

**1. Generalizations of existing methods.** We introduce the concept of a *Shifted Compressor*, which generalizes a common definition of compression operators used in distributed learning. This technique allows to study various strategies for updating the shifts using both biased and unbiased compressors, to recover and improve such previously known methods as DCGD and DIANA. Additionally, as a byproduct, a new algorithm is obtained: DCGD-STAR, which achieves linear convergence to the exact solution if we know the local gradients at the optimum.

**2. Improved rates.** The notion of a shifted compressor allows us to revisit existing analysis of distributed methods

with *compressed iterates* and improve guarantees in both cases: with and without variance-reduction. Obtained results indicate that algorithms with model compression can have the same complexity as compressed gradient methods.

**3. New algorithm.** We present a novel distributed algorithm with compression, called Randomized DIANA, with linear convergence rate to the exact optimum. It has a significantly *simpler analysis* than the original DIANA method. Via examination of its experimental performance we highlight the cases when it can outperform DIANA in practice.

## 2 GENERAL FRAMEWORK

In this section we introduce compression operators and the framework of shifted compressors.

### 2.1 STANDARD COMPRESSION

At first recall some basic definitions.

**Definition 1** (General contractive compressor). *A (possibly) randomized mapping* $\mathcal{C} : \mathbb{R}^d \to \mathbb{R}^d$ *is a* **compression operator** *(*$\mathcal{C} \in \mathbb{B}(\delta)$ *for brevity) if for some* $\delta \in (0, 1]$ *and* $\forall x \in \mathbb{R}^d$

$$\mathbf{E}\,\|\mathcal{C}(x) - x\|^2 \leq (1 - \delta)\|x\|^2,$$

*where the expectation is taken w.r.t. (possible) randomness of operator* $\mathcal{C}$.

One of the most known operators from this class is *greedy sparsification* (Top-K for $K \in \{1, \ldots, d\}$):

$$\mathcal{C}_{\texttt{Top-K}}(x) := \sum_{i=d-K+1}^{d} x_{(i)} e_{(i)},$$

where coordinates are ordered by their magnitudes so that $|x_{(1)}| \leq |x_{(2)}| \leq \cdots \leq |x_{(d)}|$, and $e_1, \ldots, e_d \in \mathbb{R}^d$ are the standard unit basis vectors. This compressor belongs to $\mathbb{B}\,(K/d)$.

**Definition 2** (Unbiased compressor). *A randomized mapping* $\mathcal{Q} : \mathbb{R}^d \to \mathbb{R}^d$ *is an* **unbiased compression operator** *(*$\mathcal{Q} \in \mathbb{U}(\omega)$ *for brevity) if for some* $\omega \geq 0$ *and* $\forall x \in \mathbb{R}^d$

$(a)\,\mathbf{E}\,\mathcal{Q}(x) = x,$       *(Unbiasedness)*

$(b)\,\mathbf{E}\,\|\mathcal{Q}(x) - x\|^2 \leq \omega\|x\|^2$    *(Bounded variance)*

*The last inequality implies that*

$$\mathbf{E}\,\|\mathcal{Q}(x)\|^2 \leq (1 + \omega)\|x\|^2. \tag{1}$$

A notable example from this class is the *random sparsification* (Rand-K for $K \in \{1, \ldots, d\}$) operator:

$$\mathcal{Q}_{\texttt{Rand-K}}(x) := \frac{d}{K} \sum_{i \in S} x_i e_i, \tag{2}$$

where $S$ is a random subset of $[d] := \{1, \ldots, d\}$ sampled from the uniform distribution on the all subsets of $[d]$ with cardinality $K$. Rand-K belongs to $\mathbb{U}\,(d/K - 1)$.

Notice that property (a) from Definition 2 is "uniform" across all vectors $x$, while property (b) is not. Namely, vector $x = 0$ is treated *in a special way* because $\mathbf{E}\,\|\mathcal{Q}(0) - 0\|^2 = 0$, which means that the compressed zero vector has *zero variance*. In other words, zero is mapped to itself with probability 1.

### 2.2 COMPRESSION WITH SHIFT

We can generalize the class of unbiased compressors $\mathbb{U}(\omega)$ to a class of operators with other (not only 0) "special" vectors. Specifically, this class allows for **shifts** away from the origin, which is formalized in the following definition.

**Definition 3** (Shifted compressor). *A randomized mapping* $\mathcal{Q}_h : \mathbb{R}^d \to \mathbb{R}^d$ *is a* **shifted compression operator** *(*$\mathcal{Q}_h \in \mathbb{U}(\omega; h)$ *in short) if exists* $\omega \geq 0$ *such that* $\forall x \in \mathbb{R}^d$

$(a)\,\mathbf{E}\,\mathcal{Q}_h(x) = x$

$(b)\,\mathbf{E}\,\|\mathcal{Q}_h(x) - x\|^2 \leq \omega\|x - h\|^2.$

*Vector* $h \in \mathbb{R}^d$ *is called a* **shift**. *Note that class of unbiased compressors* $\mathbb{U}(\omega)$ *is equivalent to* $\mathbb{U}(\omega; 0)$.

The next lemma shows that shifts add up and all shifted compression operators $\mathcal{Q}_h \in \mathbb{U}(\omega; h)$ arise by a shift of some operator $\mathcal{Q}_0$ from $\mathbb{U}(\omega; 0)$.

**Lemma 1** (Shifting a shifted compressor). *Let* $\mathcal{Q}_h \in \mathbb{U}(\omega; h)$ *and* $v \in \mathbb{R}^d$. *Then the (possibly) randomized mapping* $\mathcal{Q}$ *defined by*

$$\mathcal{Q}(x) := v + \mathcal{Q}_h(x - v)$$

*satisfies* $\mathcal{Q} \in \mathbb{U}(\omega; h + v)$.

*The shifted compressor* concept allows us to construct a shifted compressed **gradient estimator** $\mathcal{Q}_h \in \mathbb{U}(\omega; h)$ given by

$$g_h(x) = \mathcal{Q}_h\,(\nabla f(x)) = h + \mathcal{Q}(\nabla f(x) - h), \tag{3}$$

which is the main focus of this work. In particular, we are going to study different mechanisms for choosing this shift vector throughout the optimization process.

*Note:* The estimator (3) is clearly unbiased, as soon as the operator $\mathcal{Q}$ satisfies $\mathbf{E}\,\mathcal{Q}(x) = x$.

Estimator (3) uses operator $\mathcal{Q}$ from class of unbiased compressors $\mathbb{U}(\omega)$, which are usually easier to analyze but have higher empirical variance than their biased counterparts (Beznosikov et al., 2020). In an attempt to kill two birds

with one stone, we can incorporate the (possibly) biased compressor $\mathcal{C} \in \mathbb{B}(\delta)$ into $h$ using a similar shift trick:

$$h = s + \mathcal{C}(\nabla f(x) - s), \tag{4}$$

as $g_h(x)$ allows for virtually any shift vector. This leads to the following estimator[1]

$$
\boxed{
\begin{aligned}
g_h(x) &= h + \mathcal{Q}\left(\nabla f(x) - h\right) \\
&= s + \mathcal{C}(\nabla f(x) - s) \\
&\quad + \mathcal{Q}\left(\nabla f(x) - s - \mathcal{C}(\nabla f(x) - s)\right).
\end{aligned}}
\tag{5}
$$

## 2.3 THE META-ALGORITHM

Now we are ready to present the general distributed optimization algorithm for solving ($\star$) that employs shifted gradient estimators

$$g_h(x) = \frac{1}{n}\sum_{i=1}^{n} g_{h_i}(x) = \frac{1}{n}\sum_{i=1}^{n}\left[h_i + \mathcal{Q}_i\left(\nabla f_i(x) - h_i\right)\right].$$

---

**Algorithm 1** Distributed Compressed Gradient Descent with Shift (DCGD-SHIFT)

1: **Parameters:** learning rate $\gamma > 0$; unbiased compressors $\mathcal{Q}_1, \ldots, \mathcal{Q}_n$; initial iterate $x^0 \in \mathbb{R}^d$, initial local shifts $h_1^0, \ldots, h_n^0 \in \mathbb{R}^d$ (stored on the $n$ nodes)
2: **Initialize:** $h^0 = \frac{1}{n}\sum_{i=1}^{n} h_i^0$ (stored on the master)
3: **for** $k = 0, 1, 2 \ldots$ **do**
4:      Broadcast $x^k$ to all workers
5:      **for** $i = 1, \ldots n$ **do** in parallel
6:          Compute local gradient: $\nabla f_i(x^k)$
7:          Compress: $m_i^k = \mathcal{Q}_i(\nabla f_i(x^k) - h_i^k)$
8:          Update the local shift: $h_i^{k+1}$
9:          Send $m_i^k$ and/or (maybe) $h_i^{k+1}$ to the master
10:      **end for**
11:      Aggregate received messages: $m^k = \frac{1}{n}\sum_{i=1}^{n} m_i^k$
12:      Compute global estimator: $g^k = h^k + m^k$
13:      Take gradient descent step: $x^{k+1} = x^k - \gamma g^k$
14:      Update aggregated shift: $h^{k+1} = \frac{1}{n}\sum_{i=1}^{n} h_i^{k+1}$
15: **end for**

---

In Algorithm 1, each worker $i = 1, \ldots, n$ queries the gradient oracle $\nabla f_i(x^k)$ in iteration $k$. Then, a compression operator is applied to the difference between the local gradient and shift, and the result is sent to the master (and also possibly the new shift). The shift is updated on both the server and workers. After receiving the messages $m_i^k$, a

---

[1] The resulting estimator is related to induced compressor (Horváth and Richtárik, 2021) $\mathcal{Q}_{ind}(x) = \mathcal{C}(x) + \mathcal{Q}(x - \mathcal{C}(x))$, which belongs to the $\mathbb{U}(\omega(1 - \delta))$ class for $\mathcal{C} \in \mathbb{B}(\delta)$ and $\mathcal{Q} \in \mathbb{U}(\omega)$.

global gradient estimator $g^k$ is formed on the server, and a gradient step is performed.

Note that this method is not fully defined because it requires a description of the mechanism for updating the shifts $h_i^{k+1}$ (highlighted in color) throughout the iteration process on both workers and master. In the next section, we illustrate how the shifts can be chosen and updated.

## 3 CHOOSING THE SHIFTS

First, in Table 2, we show the generality of our approach by presenting some of the existing and new distributed methods that fall into our framework of DCGD-SHIFT with shift updates of the form (4).

The following assumptions are needed to analyze convergence and compare with previous results.

**Assumption 1** (Strong convexity). *Function $f : \mathbb{R}^d \to \mathbb{R}$ is $\mu$-strongly convex if*

$$f(x) \geq f(y) + \langle \nabla f(y), x - y\rangle + \frac{\mu}{2}\|x - y\|^2, \ \forall x, y \in \mathbb{R}^d.$$

*If $\mu = 0$, then the function is convex.*

**Assumption 2** (Smoothness). *Function $f : \mathbb{R}^d \to \mathbb{R}$ is $L$-smooth if*

$$f(x) \leq f(y) + \langle \nabla f(y), x - y\rangle + \frac{L}{2}\|x - y\|^2, \ \forall x, y \in \mathbb{R}^d.$$

Now, we can provide a general convergence guarantee for Algorithm 1 with fixed shifts

$$h_i^k \equiv h_i. \tag{6}$$

**Theorem 1** (DCGD with fixed SHIFT). *Assume each $f_i$ is convex and $L_i$-smooth, and $f$ is $L$-smooth and $\mu$-strongly convex. Let $\mathcal{Q}_i \in \mathbb{U}(\omega_i)$ be independent unbiased compression operators. If the step-size satisfies*

$$\gamma \leq \frac{1}{L + 2\max_i\left(L_i\omega_i/n\right)},$$

*then the iterates of Algorithm 1 with fixed shifts $h_i^k \equiv h_i$ satisfy*

$$
\begin{aligned}
\mathbf{E}\left\|x^k - x^\star\right\|^2 &\leq (1 - \gamma\mu)^k\|x^0 - x^\star\|^2 \\
&\quad + \frac{2\gamma}{\mu}\frac{1}{n}\sum_{i=1}^{n}\frac{\omega_i}{n}\left\|\nabla f_i(x^\star) - h_i\right\|^2.
\end{aligned}
\tag{7}
$$

This theorem establishes a linear convergence rate up to a certain oscillation radius, controlled by the average distance of shift vectors $h_i$ to the optimal local gradients $\nabla f_i(x^\star)$ multiplied by the step-size $\gamma$. This means that in the interpolation/**overparameterized regime** ($\nabla f_i(x^\star) = 0$ for all $i$), method reaches **exact solution** with zero shifts $h_i^0 = 0$.

Table 2: List of existing and new algorithms that fit our general framework. **VR** – variance reduced method. $\mathcal{O}/\mathcal{I}$ – zero/identity operator, $\mathcal{B}_{p_i}$ – Bernoulli[2] compressor. DGD refers to Distributed Gradient Descent.

| | | | **Shift** $h_i^{k+1} = s_i^k + \mathcal{C}_i\left(\nabla f_i(x^k) - s_i^k\right)$ | |
| Method | Reference | VR | $s_i^k$ | $\mathcal{C}_i$ |
| --- | --- | --- | --- | --- |
| DCGD | (Khirirat et al., 2018) | ✗ | $0$ | $\mathcal{O}$ |
| DCGD-SHIFT | (this work) | ✗ | $s_i^0$ | $\mathcal{O}$ |
| DGD | (folklore) | ✓ | $0$ | $\mathcal{I}$ |
| DCGD-STAR | (this work) | ✓ | $\nabla f_i(x^\star)$ | any $\mathcal{C}_i \in \mathbb{B}(\delta)$ |
| DIANA | (Mishchenko et al., 2019) | ✓ | $h_i^k$ | $\alpha \mathcal{Q}_i,\ \mathcal{Q}_i \in \mathbb{U}(\omega_i)$ |
| RAND-DIANA | (this work) | ✓ | $h_i^k$ | $\mathcal{B}_{p_i}$ |
| GDCI | (Chraibi et al., 2019) | ✗ | $x^k/\gamma$ | $\mathcal{O}$ |

In the following subsections, we study how the shifts can be formed to guarantee linear convergence to the exact optimum. We start by introducing practically useless, but theoretically insightful DCGD-STAR, and then move onto implementable algorithms that learn the optimal shifts.

## 3.1  OPTIMAL SHIFTS

Assume, for the sake of argument, that we know the values $\nabla f_i(x^\star)$ for every $i \in [n]$. Then, we can construct optimally shifted compressed shift updates sequence using the form (4)

$$h_i^{k+1} = \nabla f_i(x^\star) + \mathcal{C}_i(\nabla f_i(x^k) - \nabla f_i(x^\star)). \quad (8)$$

This is enough to fully characterize the Algorithm 1 and obtain the following convergence guarantee:

**Theorem 2** (DCGD-STAR). *Assume each $f_i$ is convex and $L_i$-smooth, and $f$ is $L$-smooth and $\mu$-strongly convex. Let $\mathcal{Q}_i \in \mathbb{U}(\omega_i), \mathcal{C}_i \in \mathbb{U}(\delta_i)$ be independent compression operators. If the step-size satisfies*

$$\gamma \leq \frac{1}{L + \max_i\left(L_i\omega_i(1-\delta_i)/n\right)}, \quad (9)$$

*then the iterates of* DCGD *with* ***optimally shifted compressed shift*** *update* (8) *satisfy*

$$\mathbf{E}\left\|x^k - x^\star\right\|^2 \leq (1-\gamma\mu)^k\|x^0 - x^\star\|^2.$$

This is the first presented algorithm with linear convergence to the exact solution for the general *not-overparameterized case*. Notice that for zero-identity operators $\mathcal{C}_i \equiv 0$ we obtain the simplest optimal shift $h_i = \nabla f_i(x^\star)$ and the term $\delta_i$ in (9) should be interpreted as zero.

The issue with the described method is that, in general, we do not know the values $h_i^\star := \nabla f_i(x^\star)$ (unless the problem

is overparametrized), which makes method impractical.

## 3.2  LEARNING THE OPTIMAL SHIFTS

We need to design the sequences $\{h_1^k\}_{k\geq 0}, \ldots, \{h_n^k\}_{k\geq 0}$ in such a way that they all converge to the optimal shifts:

$$h_i^k \to \nabla f_i(x^\star) \quad \text{as} \quad k \to \infty.$$

However, at the same time, we do not want to send uncompressed vectors from workers to the master. So, the challenge is not only learning the shifts, but doing so in a communication-efficient way. We present two different solutions to this problem in this work.

### 3.2.1  DIANA-like Trick

Our first approach is based on the celebrated DIANA (Mishchenko et al., 2019; Horváth et al., 2019b) algorithm:

$$\begin{aligned} h_i^{k+1} = h_i^k + \alpha\big[&\mathcal{C}_i(\nabla f_i(x^k) - h_i^k) \\ &+ \mathcal{Q}_i\left(\nabla f_i(x^k) - h_i^k - \mathcal{C}_i(\nabla f_i(x^k) - h_i^k)\right)\big], \end{aligned} \quad (10)$$

where $\alpha$ is a suitably chosen step-size. For $\mathcal{C}_i \equiv 0$, it takes the simplified form

$$h_i^{k+1} = h_i^k + \alpha\mathcal{Q}_i\left(\nabla f_i(x^k) - h_i^k\right). \quad (11)$$

This recursion resolves both of the raised issues earlier. Firstly, this sequence of $h_i^k$ indeed converges to the optimal shifts $\nabla f_i(x^\star)$, which is formalized in the Theorem 3 presented later. Moreover, the shift on the master

---

$$^2\mathcal{B}_p(x) := \begin{cases} x & \text{with probability } p \\ 0 & \text{with probability } 1 - p \end{cases}$$

$h^{k+1} = \frac{1}{n} \sum_{i=1}^{n} h_i^{k+1}$ is updated as follows:

$$
\begin{aligned}
h^{k+1} &= \frac{1}{n} \sum_{i=1}^{n} \Big\{ h_i^k + \alpha \big[ \mathcal{C}_i(\nabla f_i(x^k) - h_i^k) \\
&\qquad + \mathcal{Q}_i \left( \nabla f_i(x^k) - h_i^k - \mathcal{C}_i(\nabla f_i(x^k) - h_i^k) \right) \big] \Big\} \\
&= \frac{1}{n} \sum_{i=1}^{n} h_i^k + \alpha \frac{1}{n} \sum_{i=1}^{n} \left\{ c_i^k + m_i^k \right\} \\
&= h^k + \alpha \left( c^k + m^k \right),
\end{aligned}
$$

which requires aggregation of the compressed vectors $c_i^k := \mathcal{C}_i(\nabla f_i(x^k) - h_i^k)$ and $m_i^k := \mathcal{Q}_i \left( \nabla f_i(x^k) - h_i^k - c_i^k \right)$ from the workers. In the case of update (11), it is not even needed to send anything in addition to the messages $m_i^k$ required by default in Algorithm 1.

Furthermore, simplified recursion (11) can be interpreted as one step of Compressed Gradient Descent (CGD) with step-size $\alpha$ applied to such optimization problem:

$$
\max_{h_i \in \mathbb{R}^d} \left[ \phi_i^k(h_i) := -\frac{1}{2} \left\| h_i - \nabla f_i(x^k) \right\|^2 \right],
$$

which is in fact a 1-smooth and 1-strongly concave function. In this way, $h_i^{k+1}$ keeps track of the latest local gradient and produces a better estimate than the previous shift $h_i^k$.

Now we present the convergence result for the Algorithm 1 with described before shift learning procedure.

**Theorem 3** (Generalized DIANA). *Assume each $f_i$ is convex and $L_i$-smooth, and $f$ is $\mu$-strongly convex. Let $\mathcal{Q}_i \in \mathbb{U}(\omega_i), \mathcal{C}_i \in \mathbb{U}(\delta_i)$ be independent compression operators. If the step-sizes for all $i$ satisfy*

$$
\begin{aligned}
\alpha &\le \frac{1}{1 + \omega_i(1 - \delta_i)}, \\
\gamma &\le \frac{1}{\frac{2}{n} \max_i (\omega_i L_i) + (1 + \alpha M) L_{\max}},
\end{aligned}
$$

*where $L_{\max} := \max_i L_i, M > 2/(n\alpha)$ and $\delta_i$ should be interpreted as zero for $\mathcal{C}_i \equiv 0$, then the iterates of DCGD with the DIANA-like shift update (10) satisfy*

$$
\mathbf{E} \, V^k \le \max \left\{ (1 - \gamma\mu)^k, \left( 1 - \alpha + \frac{2\omega}{nM} \right)^k \right\} V^0,
$$

*where the Lyapunov function $V^k$ is defined by*

$$
V^k := \left\| x^k - x^\star \right\|^2 + M\gamma^2 \cdot \frac{1}{n} \sum_{i=1}^{n} \omega_i \left\| h_i^k - \nabla f_i(x^\star) \right\|^2.
$$

Our result represents an improvement over the original DIANA in several ways. Firstly, we use a much more general shift updates involving $\mathcal{C}_i$, which allow biased operators to be used for learning the optimal shifts. Secondly, one can use different compressors $\mathcal{Q}_i$, which can be particularly beneficial when different workers have various bandwidths/connection speeds to the master. Thus, the slower workers can compress more, and therefore use operators with higher $\omega_i$. At the same, time the opposite makes sense for "faster" workers.

### 3.2.2 Randomized DIANA (Rand-DIANA)

Recalling the original issue stated in Section 3.2 that we are dealing with:

**design sequences** $\{h_i^k\}_{k \ge 0}$ such that $h_i^k \to \nabla f_i(x^\star)$.

The simplest possible solution would be just to set $h_i^k$ to $\nabla f_i(x^k)$ because if $x^k \to x^\star$ in the optimization process, then $\nabla f_i(x^k)$ converges to the optimal local shift. However, this approach is not efficient, as workers have to transfer full (uncompressed) vectors $h_i^k = \nabla f_i(x^k)$. Our alternative to the DIANA solution is to update a reference point $w_i^k$ for calculating the shift $h_i^k = \nabla f_i(w_i^k)$ infrequently (with a small probability $p_i \in (0, 1]$), so that $h_i^k$ needs to be communicated very rarely:

$$
\begin{aligned}
h_i^k &= \nabla f_i(w_i^k) \\
w_i^{k+1} &= \begin{cases} x^k & \text{with probability } p_i \\ w_i^k & \text{with probability } 1 - p_i \end{cases}
\end{aligned} \tag{12}
$$

This method has a remarkably simpler analysis than DIANA, but can solve the original problem of eliminating the variance introduced by gradient compression. Next, we state the convergence result for DCGD with shifts updated in a `randomized` fashion (12). We named it Randomized-DIANA (Rand-DIANA in short) to acknowledge the original method (Mishchenko et al., 2019) to first solve this problem.

**Theorem 4** (Rand-DIANA). *Assume that $f_i$ are convex, $L_i$-smooth for all $i$ and $f$ is $\mu$-convex. If the step-size satisfies*

$$
\gamma \le \frac{1}{\left( 1 + \frac{2\omega}{n} \right) L_{\max} + M \max_i(p_i L_i)},
$$

*where $M > \frac{2\omega}{np_m}$ and $p_m := \min_i p_i$. Then, the iterates of DCGD with Randomized-DIANA shift update (12) satisfy*

$$
\mathbf{E} \, V^k \le \max \left\{ (1 - \gamma\mu)^k, \left( 1 - p_m + \frac{2\omega}{nM} \right)^k \right\} V^0,
$$

*where the Lyapunov function $V^k$ is defined by*

$$
V^k := \left\| x^k - x^\star \right\|^2 + M\gamma^2 \cdot \frac{1}{n} \sum_{i=1}^{n} \left\| h_i^k - \nabla f_i(x^\star) \right\|^2.
$$

Though appropriate choice of the parameters $M = \frac{4\omega}{np_m}$ and $p_i \equiv p = \frac{1}{\omega + 1}$ for every $i$, we can obtain basically the same iteration complexity as the original DIANA (Horváth et al., 2019b)

$$
\max \left\{ \frac{1}{\gamma\mu}, \frac{1}{p_m - \frac{2\omega}{nM}} \right\} = \max \left\{ \frac{L_{\max}}{\mu} \left( 1 + \frac{\omega}{n} \right), \omega + 1 \right\}.
$$

## 3.3 COMPRESSING THE ITERATES

In this section, we discuss how the shifted compression framework can be applied and leads to improved results for the case where the iterates/models themselves need to be compressed.

Let $\mathcal{Q} \in \mathbb{U}(\omega)$. Consider the following shifted by vector $x/\gamma$ compressor

$$\hat{\mathcal{Q}}(z) := \frac{x}{\gamma} + \mathcal{Q}\left(z - \frac{x}{\gamma}\right),$$

which clearly belongs to the class $\mathbb{U}(\omega; x/\gamma)$. Based on the fact that for $\gamma \neq 0$ compressor $\bar{\mathcal{Q}}(z) := -\frac{1}{\gamma} \cdot \mathcal{Q}(-\gamma z) \in \mathbb{U}(\omega)$ we can transform $\hat{\mathcal{Q}}$ to operator

$$\tilde{\mathcal{Q}}(z) := \frac{x}{\gamma} + \bar{\mathcal{Q}}\left(z - \frac{x}{\gamma}\right) = \frac{1}{\gamma}\left[x - \mathcal{Q}(x - \gamma z)\right],$$

which also belongs to $\mathbb{U}(\omega; x/\gamma)$ and is helpful for analysing algorithms with compressed iterates.

**Distributed Gradient Descent with Compressed Iterates** (GDCI) was first analyzed by Khaled and Richtárik (2019) for single node and, in short, was relaxed and formulated in a convenient form by Chraibi et al. (2019):

$$x^{k+1} = (1-\eta)x^k + \eta\mathcal{Q}\left(x^k - \gamma\nabla f(x^k)\right). \quad \text{(GDCI)}$$

This algorithm can be reformulated using the previously described shifted compressor $\tilde{\mathcal{Q}} \in \mathbb{U}(\omega; x^k/\gamma)$

$$x^{k+1} = x^k - (\eta\gamma)\frac{1}{\gamma}\left[x^k - \mathcal{Q}\left(x^k - \gamma\nabla f(x^k)\right)\right]$$
$$= x^k - (\eta\gamma)\tilde{\mathcal{Q}}^k(\nabla f(x^k)),$$

which for the distributed case takes the form

$$x^{k+1} = (1-\eta)x^k + \eta\frac{1}{n}\sum_{i=1}^{n}\mathcal{Q}_i\left(x^k - \gamma\nabla f_i(x^k)\right). \quad (13)$$

The essence of this method is compression of the local workers' iterates $\mathcal{Q}_i\left(x^k - \gamma\nabla f_i(x^k)\right)$, their aggregation on the master and convex combination with the previous model. Next we present established linear convergence up to a neighborhood introduced due to variance of compression operator (similarly to DCGD with fixed shifts Theorem 1).

**Theorem 5** (GDCI). *Assume each $f_i$ is convex and $L_i$-smooth, and $f$ is $L$-smooth and $\mu$-strongly convex. Let $\mathcal{Q}_i \in \mathbb{U}(\omega)$ be independent compression operators. If the step-sizes satisfy*

$$\eta \leq \left[\frac{L}{\mu} + \frac{2\omega}{n}\left(\frac{L_{\max}}{\mu} - 1\right)\right]^{-1}, \gamma \leq \frac{1 + 2\eta\omega/n}{\eta\left(L + 2L_{\max}\omega/n\right)},$$

*then the iterates of the Distributed GDCI (13) satisfy*

$$\mathbf{E}\left\|x^k - x^\star\right\|^2 \leq (1-\eta)^k\|x^0 - x^\star\|^2$$
$$+ \eta\frac{2\omega}{n}\frac{1}{n}\sum_{i=1}^{n}\|x^\star - \gamma\nabla f_i(x^\star)\|^2. \quad (14)$$

In the interpolation regime ($\nabla f_i(x^\star) = 0 = x^\star - \gamma\nabla f_i(x^\star)$, for every $i$) this result matches the complexity of DCGD with fixed shifts (7)

$$\tilde{\mathcal{O}}\left(\kappa\left(1 + \omega/n\right)\right)$$

and improves over the original rate of GDCI by Chraibi et al. (2019) analyzed for fixed point problems and specialized for gradient mappings:

$$\tilde{\mathcal{O}}\left(\kappa\max\left\{1, \kappa\omega/n\right\}\right) \gtrsim \tilde{\mathcal{O}}\left(\kappa^2\omega/n\right).$$

Due to space limitations, the results for **Distributed Variance-Reduced Gradient Descent with Compressed Iterates** (VR-GDCI), which eliminates the neighborhood in (14), along with detailed proofs of all stated theorems are presented in the Supplementary Material.

## 4 EXPERIMENTS

In this section, we present some of the experimental results obtained. The remainder of the results (including real-world data and other models) are available in the Supplementary Materia. To provide evidence that our theory translates into observable predictions, we focus on well-controlled settings that satisfy the assumptions in our work.

Consider a classical ridge-regression optimization problem

$$\min_{x\in\mathbb{R}^d}\left[f(x) := \frac{1}{2}\|\mathrm{A}x - y\|^2 + \frac{\lambda}{2}\|x\|^2\right],$$

where $\lambda = 1/m$ and $\mathrm{A} \in \mathbb{R}^{m\times d}, y \in \mathbb{R}^m$ are generated using the Scikit-learn library (Pedregosa et al., 2011) method sklearn.datasets.make_regression with default parameters for $m = 100, d = 80$. The obtained data is uniformly, evenly, and randomly distributed among 10 workers. To compare selected algorithms, we evaluate the logarithm of a relative argument error $\log\left(\|x^k - x^\star\|^2/\|x^0 - x^\star\|^2\right)$ on the vertical axis, while the horizontal axis presents the number of communicated bits needed to reach a certain error tolerance $\varepsilon$. The starting point $x^0 \in \mathbb{R}^d$ entries are sampled from the normal distribution $\mathcal{N}(0, 10)$.

In our simulations we thoroughly examine the Rand-DIANA method, which is presented for the first time. Extensive studies of the methods with compressed iterates can be found in the works by Khaled and Richtárik (2019); Chraibi et al. (2019).

### 4.1 RANDOMIZED-DIANA VS DIANA

In the first set of experiments, we compare Rand-DIANA and DIANA with different compressors $\mathcal{Q}_i$ ($\mathcal{C}_i \equiv 0$) and varied operators' parameters. The results obtained are summarized in Figure 1. The designation $q := k/d$ is used for the share of non-zeroed coordinates of the

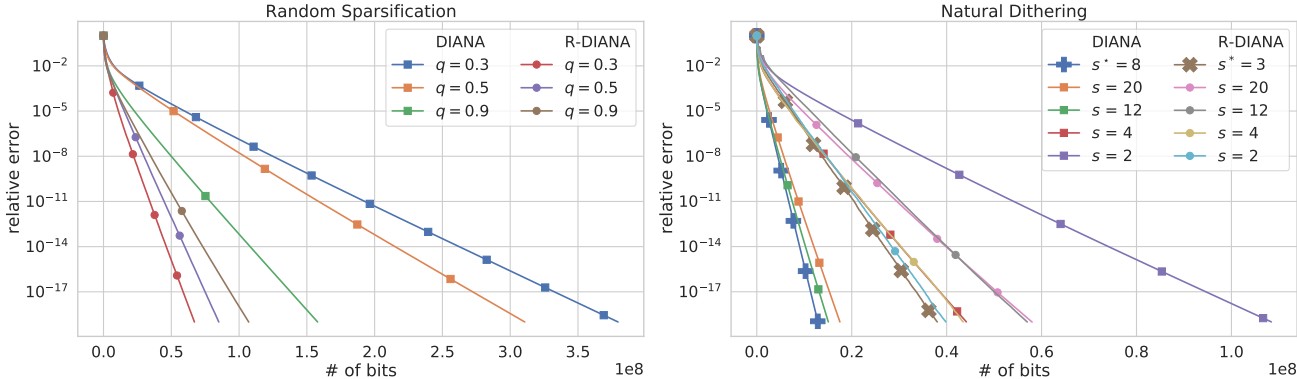

Figure 1: Comparison of DIANA and Randomized-DIANA. **Left plot**: methods equipped with `Rand-K` for different $q$ values. **Right plot**: selected results of a grid search for the `ND` parameter $s$ over $\{2, \ldots, 20\}$.

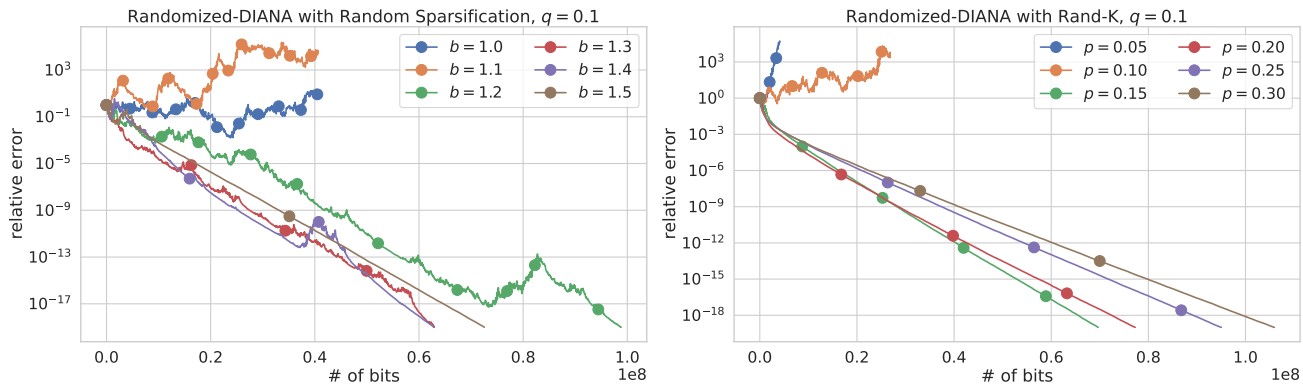

Figure 2: Study of the stability and performance of Rand-DIANA with varying parameters $b$ and $p$.

Random sparsification (`Rand-K`) operator, and $s$ corresponds to the number of levels for the `Natural Dithering` (`ND`) (Horváth et al., 2019a) compressor. The $p$ parameter of Rand-DIANA was set at $1/(\omega + 1)$ for every run.

The left plot in Figure 1 clearly shows that Rand-DIANA performs better than DIANA for every value of the `Rand-K` compressor parameter. It is worth noting that DIANA performs better at higher $q$, while the opposite holds for Rand-DIANA.

From the right plot in Figure 1, one can see that DIANA with `ND` can be superior to Rand-DIANA for the optimized parameter $s^\star$. Nevertheless, Rand-DIANA is highly preferable for very aggressive compression (e.g., $s = 2$).

In the next experimental setup, we more closely investigate the behavior of Rand-DIANA with respect to its parameters.

### 4.2 RANDOMIZED-DIANA **STUDY**

According to the formulation of Theorem 4, the constant $M$ has to be strictly greater than $M' \coloneqq 2\omega/(np)$. In the left plot of Figure 2, we show that the method becomes less stable and can even diverge for smaller values of $M$ (set to $M' \cdot b$). However, too high $M$ (for $b = 1.5$) can lead to an overall (stable) slowdown. We conclude that the condition imposed by theoretical analysis is indeed critical.

The right plot in Figure 2 examines how the parameter $p$ affects the convergence in a high compression regime ($q = 0.1$). The method converges faster for smaller $p$ and can diverge above a certain threshold, similarly to the previous study of $M$ trade-off.

We did not conduct additional experiments to show the effect of combining unbiased compressors with biased counterparts, as the benefits of such an approach have already been clearly demonstrated by Horváth and Richtárik (2021) for distributed training of deep neural networks.

### Acknowledgements

We would like to thank the anonymous reviewers, Laurent Condat and Konstantin Mishchenko for their helpful comments and suggestions to improve the manuscript.

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

pages 1 and 2)
