# OpenReview forum: "Shifted Compression Framework: Generalizations and Improvements"
_auai.org/UAI/2022/Conference — UAI 2022 Poster_

### Official Review · Reviewer_SQqX · 2022-03-23

**Q2(1) Originality/Novelty:** 2
**Q2(2) Significance/Impact:** 2
**Q2(3) Correctness/Technical Quality:** 3
**Q2(6) Clarity Of Writing:** 3
**Q6 Overall Score:** 4
**Q8 Confidence In Your Score:** 3

**Q1 Summary And Contributions:**

This paper proposes novel shifted compressors along with a general distributed optimization framework. The proposed framework requires communicating both the compressed value and the shifts. To reduce the communication overhead of the shifts, a randomized DIANA method is used. The theoretical analysis shows that applying shifted compressors to some existing distributed optimization algorithms can result in better convergence. The experiments show good performance.

**Q2 Assessment Of The Paper:**

More detailed information regarding each of these aspects is given below:

**Q2(4) Quality Of Experiments (Optional):**

2: Fair: The experimental evaluation is weak: important baselines are missing, or the results do not adequately support the main claims.

**Q2(5) Reproducibility:**

4: Excellent: Key resources (e.g., proofs, code, data) are available and key details (e.g., proof sketches, experimental setup) are comprehensively described for competent researchers to confidently and easily reproduce the main results.

**Q3 Main Strengths:**

1. The idea of shifted compression itself is neat and clean, and could be easily applied to a lot of existing distributed optimization algorithms.
2. The theoretical analysis shows that applying shifted compressors to some existing distributed optimization algorithms such as GDCI can result in better convergence.
3. The experiments on ridge regression and logistic regression show good performance compared to DIANA.

**Q4 Main Weakness:**

1. It is unclear why using DIANA in Rand-DIANA to compress the communication of the shifts. Actually, I cannot even understand how DIANA is used in Rand-DIANA, since Rand-DIANA seems simply a random masking.
2. My main concern is that the experiments are too simple. Both the optimization problems (ridge regression, logistic regression) and the datasets (synthetic, w2a) are very simple and small for modern computation hardwares. These problems could be easily and quickly solved on a single-node CPU machine, which could hardly used to justify the results for distributed optimization algorithms. Distributed training is only necessary when the model and the datasets are both extremely large.
3. The experiments only show relative error vs. number of bits. For distributed training, what the users really care is whether the overall training time could be reduced. However, the wall-clock time for the training are not reported.
4. The uncompressed baseline is not reported in the experiments.

**Q5 Detailed Comments To The Authors:**

1. In the experiments, is the algorithm called "DIANA" the DIANA itself, or Algorithm 1 with DIANA on the shifts?
2. How is DIANA used in Rand-DIANA? I cannot see any connection between DIANA and Rand-DIANA, since Rand-DIANA seems simply a random masking in the communication.
3. For distributed training, what the users really care is whether the overall training time could be reduced. I strongly recommend to report the wall-clock time for the training, and compare it with the baseline (including uncompressed training baseline).
3. I strongly recommend to report the uncompressed baseline in the experiments, so that the reader can see the gap between the compressed training and the uncompressed training.

**Q7 Justification For Your Score:**

The overall idea is technically sound. The theoretical analysis is good. However, the experiments are too simple and lack some important results and baseline. Thus, in overall, I think this paper is not ready to get published.

**Q9 Complying With Reviewing Instructions:**

1: Yes.

---

### Official Review · Reviewer_oon4 · 2022-04-14

**Q2(1) Originality/Novelty:** 2
**Q2(2) Significance/Impact:** 3
**Q2(3) Correctness/Technical Quality:** 3
**Q2(6) Clarity Of Writing:** 3
**Q6 Overall Score:** 5
**Q8 Confidence In Your Score:** 4

**Q1 Summary And Contributions:**

This paper mainly studies communication compression strategies and develops a unified framework for studying gradient compression strategies. Specifically, This paper introduces the concept of shift compressor, provides a generalization of existing methods, and improves the concept of shift compression to new algorithms.

**Q2 Assessment Of The Paper:**

More detailed information regarding each of these aspects is given below:

**Q2(4) Quality Of Experiments (Optional):**

2: Fair: The experimental evaluation is weak: important baselines are missing, or the results do not adequately support the main claims.

**Q2(5) Reproducibility:**

3: Good: Key resources (e.g., proofs, code, data) are available and key details (e.g., proofs, experimental setup) are sufficiently well-described for competent researchers to confidently reproduce the main results.

**Q3 Main Strengths:**

1. The paper widely covers several distributed communication compression algorithms, and combines the existing work with the proposed shift compressor. It provides a new aspect of analyzing compression algorithms.
2. This paper has a clear structure, with sufficient discussion of related work and several comparisons.


**Q4 Main Weakness:**

see detailed comments

**Q5 Detailed Comments To The Authors:**

1. The experimental results seem a little bit weak. The authors just provided the classical ridge regression optimization problem, which is quite poor to show the outperformance of the proposed methods. Although the authors mentioned that the benefits have been clearly demonstrated by other related work, it is better to show results on the common deep learning baselines in this paper.
2. I am also a bit concerned on the novelty of the shift compressor.  In particular, I feel this is quite similar to the EF21 compressor proposed in

     *"EF21: A new, simpler, theoretically better, and practically faster error feedback." Advances in Neural Information Processing Systems 34 (2021).*

     of course, this paper provides a general view of such compressors. Yet it still seems highly related and the intuition seems quite similar. The authors might want to claim the differences with the above work.

3. The summary in the contribution part claimed about “Improved rates”. Yet the authors mentioned in the following, “the results … can have the same complexity as compressed gradient methods”. It seems from this claim the shifted compressor maintains the convergence rate instead of really “improving” it?
4. The authors can explain more about the shift compressor and the meta-algorithm. The eq.(3) shows that it subtracts a shift $h$, compresses and then adds back the shift $h$, it involves two more steps, adding more computational costs, but it is not clear about the advantage of such a shifted compressor through the theoretical analysis.
5. It seems that the choice of shifts is crucial for the proposed shift compressor. However, one may not know which shift is optimal.
6. The theory is all about convex cases, I wonder if the authors could extend it into nonconvex cases. Also, it seems to only work under gradient descent, can it be extended to the stochastic gradient case? These would largely improve the contribution of the proposed work.

minior:
There is a typo in the right plot of Figure 1, the first legend of R-DIANA should be s = 8.


**Q7 Justification For Your Score:**

I read the full paper and roughly went over the proofs. The main strengths are based on the structure of the paper, the related works and the comparisons of theoretical results. The main weaknesses are based on the experiments and the methods part of the paper. I'd love to raise my score if the authors could address my concerns above.

**Q9 Complying With Reviewing Instructions:**

1: Yes.

---

### Official Review · Reviewer_K1tV · 2022-04-23

**Q2(1) Originality/Novelty:** 2
**Q2(2) Significance/Impact:** 2
**Q2(3) Correctness/Technical Quality:** 3
**Q2(6) Clarity Of Writing:** 4
**Q6 Overall Score:** 6
**Q8 Confidence In Your Score:** 3

**Q1 Summary And Contributions:**

The paper studies distributed/federate optimization problem, with the focus of providing a general framework of analyzing compression algorithm. In a distributed optimization problem, each user holds a single piece of data $x_i$, and the goal is for a central server to optimize the objective function $\min_{w}\sum_{i\in [n]}f(w, x_i)$, and one challenge in this field is to use limited communication as transmitting the full gradient could be expensive.





**Q2 Assessment Of The Paper:**

More detailed information regarding each of these aspects is given below:

**Q2(4) Quality Of Experiments (Optional):**

2: Fair: The experimental evaluation is weak: important baselines are missing, or the results do not adequately support the main claims.

**Q2(5) Reproducibility:**

3: Good: Key resources (e.g., proofs, code, data) are available and key details (e.g., proofs, experimental setup) are sufficiently well-described for competent researchers to confidently reproduce the main results.

**Q3 Main Strengths:**

The major result of this paper is to provide a general framework for analyzing compressed gradient descent: an algorithm that has unbiased gradient and has bounded variance with respect to a *shift* vector. This general framework allow one to recover several existing algorithm, and provide some improvement. In particular, it improves the previous rate of $\max {\kappa(1+\frac{\omega}{n}), \omega }$ to $\kappa(1 + \frac{\omega}{n})$.

**Q4 Main Weakness:**

There are no major weakness of this paper (provided I do not miss important literature). One minor issue I can think of is that the algorithm only works for strong convex function and smooth, which is relative narrow. If there are simple adaptations to other setting (as usual in optimization literature), please explicitly point out in the paper.

**Q5 Detailed Comments To The Authors:**

The writing in general is good, with a few minor issue:

1. Page 5 "Our first approach is based on the celebrated DIANA...", it is unclear to me why DIANA is a celebrated one, please consider rewording.
2. There is no explanation on the parameter $\omega$, how large it could be in practical, and how people view it in literature?


**Q7 Justification For Your Score:**

The paper gives a general framework for analyzing compression based distribution gradient descent algorithm. The idea seems natural, but could be a good summary on existing work, and also it slightly improves the previous result. Hence, I tend to accept the paper.

**Q9 Complying With Reviewing Instructions:**

1: Yes.

---

### Decision · Program_Chairs · 2022-05-15

**Decision:**

Accept (Poster)

**Comment:**

Meta Review: This paper studies a distributed/federated optimization setting, with a focus on communication as the bottleneck (specifically, they focus on the case where compressed, noisy estimates of the gradient are being used, in a suitable formalization). The paper's most significant contributions are theoretical. First, they provide a framework which generalizes several extant algorithms for such settings (namely, access to an unbiased gradient and bounded variance with respect to a shift vector). Second, they provide a convergence rate analysis for their framework, along with ways to pick optimal shifts which gives an improved algorithm.
Overall, the paper makes nice theoretical contributions, though several reviewers noted the empirical evaluation is somewhat restricted (and the settings apply to the convex setting only, which might be restrictive for some practical applications).